# Design and Implementation of a CMOS-MEMS Out-of-Plane Detection Gyroscope

**DOI:** 10.3390/mi15121484

**Published:** 2024-12-10

**Authors:** Huimin Tian, Zihan Zhang, Li Liu, Wenqiang Wei, Huiliang Cao

**Affiliations:** 1Key Laboratory of Instrumentation Science & Dynamic Measurement, Ministry of Education, North University of China, Taiyuan 030051, China; tianhuimin37@163.com (H.T.); wenqiang1997@126.com (W.W.); 2School of Information and Software Engineering, University of Electronic Science and Technology of China, Chengdu 610054, China; 2021090919020@std.uestc.edu.cn; 3Mechanized Infantry Reconnaissance Department, The Army Infantry Academy of PLA, Shijiazhuang 050083, China; liuli11280713@163.com

**Keywords:** out-of-plane gyro, chip testing, finite element analysis, CMOS-MEMS

## Abstract

A MEMS gyroscope is a critical sensor in attitude control platforms and inertial navigation systems, which has the advantages of small size, light weight, low energy consumption, high reliability and strong anti-interference capability. This paper presents the design, simulation and fabrication of a Y-axis gyroscope with out-of-plane detection developed using CMOS-MEMS technology. The structural dimensions of the gyroscope were optimized through a multi-objective genetic algorithm, and modal, harmonic response and range simulation analyses were carried out to verify the reasonableness of the design. The chip measured 1.2 mm × 1.3 mm. The simulation results indicate that the driving and detecting frequencies of the gyroscope were 9215.5 Hz and 9243.5 Hz, respectively; the Q-factors were 83,790 and 46,085; the mechanical sensitivity was 4.87 × 10^−11^ m/°/s; and the operational range was ±600°/s. Chip testing shows that the static capacitance was consistent with the preset value. The error between the measured frequency characteristics and the simulation results was 1.9%. This design establishes a foundation for the integration of the gyroscope’s structure and circuitry.

## 1. Introduction

Micro Electro Mechanical System (MEMS) technology combines miniaturization, integration, intelligence and high reliability [1,2]. In recent years, with the continuous advancements in MEMS, its applications have expanded across various fields, including aerospace, military, transportation and healthcare [3]. In the field of inertial instrumentation, traditional mechanical components are increasingly being replaced by MEMS inertial devices. MEMS inertial devices can meet the demands of modern navigation, guidance and control systems by virtue of the advantages of small size, light weight, low energy consumption, high reliability and strong anti-interference ability [4,5].

Traditional MEMS gyroscopes require separate fabrication of the sensors and control circuits, followed by integration through packaging technology, which increases the system size and complexity [6,7]. In recent years, the CMOS-MEMS process, which combines complementary metal–oxide–semiconductor (CMOS) technology with MEMS technology, has enabled the integration of CMOS circuits and MEMS devices on a single chip, significantly reducing packaging costs [8]. This process merges the advantages of bulk micromachining and surface micromachining, facilitating mass production and improving system integration, while maintaining low parasitic capacitance and compact dimensions [9]. Additionally, CMOS-MEMS technology ensures flexible internal wiring and offers advantages such as variable structural thicknesses, low parasitic capacitance, three-dimensional electrostatic actuation and capacitive sensing [10].

In 2002, Hao Luo and colleagues designed a Z-axis MEMS gyroscope using the UMC 0.18 μm copper CMOS-MEMS process, which operated at 1 atm of environmental pressure without relying on Q-factor enhancement [11]. This work highlighted the advantages of copper CMOS-MEMS, including a higher mass density and low stress. The device measured 410 μm × 330 μm, with a resonance frequency of 8.8 kHz in drive mode and a sensitivity of 0.8 pV/°/s. In 2001, Xie Huikai and colleagues reported a laterally vibrating film gyroscope compatible with standard CMOS processes, utilizing an integrated comb drive for out-of-plane actuation [12]. The packaged gyroscope operated at atmospheric pressure, with a resonance frequency of 4.2 kHz in drive mode and a sensitivity of 0.12 mV/°/s. In 2003, Xie Huikai and colleagues designed a gyroscope using interconnected metal layers as an etching mask for the structure [13]. Manufactured using post-CMOS micromachining, this gyroscope combined a 1.8 μm thick film structure and a 60 μm thick bulk silicon structure, which improved the rigidity and detection accuracy. The characteristic frequency in drive mode was 5.3 kHz. In 2009, Shih-Wei Lai and colleagues designed a biaxial gyroscope using the TSMC 0.18 μm 1P6M process, with angular velocity sensitivities of 0.087 mV/°/s in the X-axis direction and 0.017 mV/°/s in the Y-axis direction [14]. In 2017, Cheng Yu Ho proposed a new CMOS-MEMS gyroscope design based on pure oxide and symmetric metal–oxide films [15]. The device occupied an area of 500 μm × 470 μm, with resonance frequencies of 5.7 kHz and 5.9 kHz in drive and detection modes, respectively, and an angular rate sensitivity of 4 μV/°/s. Vacuum sealing further enhanced the performance. Most gyroscopes fabricated using CMOS-MEMS processes employ film structures, with resonators consisting solely of metal–oxide layers. While the fabrication steps are simple and the structural release is easy, the stiffness is generally low, and the film structure is prone to warping due to residual stress. Moreover, the gyroscopes are lightweight and easily affected by air damping. This paper proposes introducing a silicon layer for resonator fabrication to prevent structural warping, enhance device flatness and improve gyroscope stability. Furthermore, to increase the mechanical sensitivity of the gyroscope, this study performed modal matching of the drive and detection frequencies. Due to the varying thickness of the designed beams, the top metal layer needed to be sacrificed as a mask, and a process flow was developed for structural release.

This paper presents the development and evaluation of a Y-axis gyroscope based on a CMOS-MEMS process, utilizing in-plane drive and out-of-plane detection methods. Section 2 describes the structural design of the gyroscope, establishes the dynamics model of this gyroscope and optimizes the parameters of the resonant beam based on a multi-objective genetic algorithm. Section 3 describes the simulation of the gyroscope, verifies the optimization results through modal simulation, and performs harmonic response and range simulation analyses. Section 4 describes the fabrication of the gyroscope, which was processed by CMOS-MEMS process, where the detection beam’s structure was processed through the top metal as a mask, and the other structures were etched through the second metal as a mask.

## 2. Gyro Design and Optimization

### 2.1. Gyro Structure Design

The operating principle of the MEMS gyroscope is based on the energy conversion between two vibration modes—drive and detection—induced by the Coriolis force [16]. The design is mainly decoupled by the beam and the frame structure. Mass blocks connected to the frame or to the substrate via a beam with a single degree of freedom are the least likely to cause cross-coupling between the actuated and detected modes. According to the structural form, gyroscopes can be classified into internally sensed, externally driven (ISOD) and internally driven, externally sensed (IDOS) [14]. The ISOD type typically offers a larger available drive comb space compared with the IDOS type, as the structural dimensions of the internal mass block are significantly smaller than those of the externally driven frame.

As shown in Figure 1, this study designed a gyroscope for out-of-plane detection, with an overall ISOD-type structure to detect the angular velocity input in the Y-axis. The driving frame in the structure was connected to the anchor point frame through U-shaped beams, which were much longer than wide and provide degrees of freedom only in the X-axis direction. In this study, the drive comb capacitor was driven by a bilateral push–pull method, where the gyro realized vibration along the length of the forked comb (X-axis direction) under electrostatic force when different voltages were applied to the drive electrodes. The electrostatic force was proportional to the amplitude of the applied DC and AC components, and the electrostatic driving force could be adjusted by changing the driving voltage with the structural parameters determined.

As shown in Figure 2, when the Y-axis angular velocity was input, the Z-axis component of the mass block motion drove the detection mass block to make an out-of-plane simple harmonic motion due to the Coriolis effect. And the amplitude of vibration was proportional to the angular velocity. The displacement of this vibration was converted into a capacitive signal by detecting the comb capacitance, and then the physical quantity that characterized the input angular velocity information was extracted by the subsequent circuit.

The horizontal axis drive and vertical axis detection were achieved through multiple metal layers within the comb structure. As shown in Figure 3a, the metal layers of the fixed combs were electrically connected to the corresponding metal layers on the moving combs, which ensured that the CMOS combs functioned comparably to polysilicon combs. The number of metal layers affected the capacitance range, and using more metal layers increased the comb thickness, which expanded the capacitance detection area, and thus, enhanced the detection signal and improved the gyro’s sensitivity [17]. The M1–M4 layers were connected through vias and served as the driving combs. The differential drive design between the moving comb and the fixed combs on the left and right sides improved the driving efficiency and interference resistance. Opposite voltages were applied to the opposing combs to generate a greater net driving force. To prevent electrode curling due to residual stresses during the manufacturing process, a silicon base was integrated into the comb structure as a supporting element.

If all the metal layers of the moving comb are electrically connected, and metals 1 and 2, as well as metals 3 and 4 in the stator, are connected separately as shown in Figure 3b, two sidewall capacitors, C1 and C2, will form. When the moving comb is displaced upward or downward by an external force, C1 and C2 will change in opposite directions, allowing the design to function as a differential signal for detecting Z-axis motion. To avoid a large DC offset, the modulation voltages should remain balanced during operation. Any resulting DC offset can be compensated for the flexible wiring capabilities of the CMOS-MEMS process [18].

### 2.2. Theoretical Analysis

#### 2.2.1. Dynamical Equation

The physical model of this gyroscope is shown in Figure 4:

The out-of-plane detected Y-axis gyroscope has degrees of freedom only in the X- and Z-directions. The motion of the two modes can be described by the following dynamical model:(1)$$\left\{\begin{array}{l} m\ddot{x}+c_{x}\dot{x}+k_{x}x=F_{dx}\\ m\ddot{z}+c_{z}\dot{z}+k_{z}z=-2m\Omega_{y}\dot{x} \end{array}\right.$$
where *m* represents the mass; *x*, *y* and *z* represent the displacements; *k_x_* and *k_z_* represent the stiffness coefficients; *c_x_* and *c_y_* represent the damping coefficients; Ω*_y_* represents the angular velocity input; and *F_d_* represents the electrostatic force. The mechanical sensitivity of the silicon micromechanical gyroscope structure can be obtained by further simplifying:(2)$$S_{\mathrm{machanical}\;}=\frac{A_{z}}{\Omega_{y}}=\frac{-2F_{d}Q_{x}}{m\omega_{d}\sqrt{{\left(\omega_{z}^{2}-\omega_{d}^{2}\right)}^{2}+\frac{\omega_{z}^{2}\omega_{d}^{2}}{Q_{z}^{2}}}}\approx \frac{-F_{d}Q_{x}}{m\omega_{d}^{2}\left(\omega_{z}-\omega_{d}\right)}=\frac{-A_{x}}{\Delta \omega }$$
where *ω_x_*, *ω_z_*, *Q_x_* and *Q_z_* are the resonant angular frequency and quality factor of the driving and detecting modes, respectively.
(3)$$\omega_{x}=\sqrt{\frac{k_{x}}{m}},\omega_{z}=\sqrt{\frac{k_{z}}{m}},Q_{x}=\frac{m\omega_{x}}{c_{x}},Q_{z}=\frac{m\omega_{z}}{c_{z}}$$

Δ*ω* is the difference between the angular frequencies of the driving and detecting modes. The resonant frequencies of the drive and detection modes should be matched to achieve a higher sensitivity. However, due to manufacturing defects and environmental variations, the two vibration frequencies may not match.

#### 2.2.2. Design of Supporting Elastic Beams

The structural configuration of the elastic beam in a micro-mechanical gyroscope is closely related to the performance parameters of the entire device, including the driving frequency, detection frequency and sensitivity. Two structural forms of elastic beams are presented in this paper, as shown in Figure 5:

The straight beam structure is the simplest configuration, with one end fixed to the frame and the free end connected to a mass block. The width of the straight beam along the x-axis is w2, the length along the y-axis is L2 and the thickness along the z-axis is h2. The stiffness of the straight beam in the Z-axis direction is defined as follows:(4)$$K_{z}=E\frac{w_{2}{h_{2}}^{3}}{{L_{2}}^{3}}$$
where *E* is the modulus of elasticity with a magnitude of 169 GPa. Straight beams have a simple structure and are easier to machine.

The structure of the U-shaped beam is similar to that of a straight beam and can be considered as the parallel configuration of two straight beams. The stiffness in the x-direction can also be regarded as the result of the parallel combination of two straight beams.

Where Kx is the straight beam stiffness, if the length of the U-shaped beam connecting block is neglected, the stiffness of the U-shaped beam in the x-direction is
(5)$$K_{Ux}=E\frac{{w_{1}}^{3}h_{1}}{2{L_{1}}^{3}}$$

The structural form can effectively eliminate the residual stress generated in the process and suppress the unstable change in the resonance frequency; the symmetry of the structure in the form itself can effectively reduce the impact of dimensional errors in the process and reduce the coupling between the driving and detecting modes.

### 2.3. Multi-Objective Genetic Algorithm Optimization

The mechanical tuning of a gyro is achieved by changing the structural parameters of the gyro to adjust its resonant frequency so that the drive frequency matches or is close to the detection frequency. This adjustment enhances the gyroscope’s response to the angular velocity, thereby improving its sensitivity [19]. The resonant frequency can be modified by increasing or decreasing the mass of the vibrating block or by adjusting the dimensions of the primary beams to alter its stiffness.

The NSGA-II (Non-dominated Sorting Genetic Algorithm II) is a commonly used multi-objective optimization algorithm that handles multiple conflicting objectives and generates a solution set of Pareto frontiers, helping the designer to find the optimal trade-off between multiple objectives [20]. The NSGA-II employs techniques such as non-dominated sorting and crowding distance, among other techniques, to provide advantages such as fast convergence, maintaining population diversity and optimization robustness [21]. Limited by the process, and especially by the thickness of the metal-oxide layer, the operating frequency of the gyro designed in this study was set to approximately 9 kHz.

A multi-objective genetic algorithm (NSGA-II) was employed to optimize the beam widths of the gyro structure, with the aim to achieve the desired eigenfrequency. The widths and lengths of the two types of beams were treated as geometrically variable parameters, where the first- and second-order frequencies, along with the displacements, served as the objective functions. By parametrically scanning the simulation design points, the relationship between the sensitivity to the operating frequency and the beam width was determined, as shown in Figure 6. In the figure, F1 and F2 represent the first-order and second-order modal frequencies, respectively, and D1 and D2 represent the first-order modal drive frame displacement and the second-order modal detection mass block displacement, respectively. It is important to note that the displacements here only characterize the shapes of the modes and cannot represent the amplitude of any physical vibration. If the size of the actual vibration mode needs to be determined, it still needs to be determined by applying excitation and damping. A higher rectangle in the graph indicates that the objective function is more influenced by this parameter. By optimizing these parameters, the eigenfrequency and frequency response can be accurately controlled [22].

As shown in Figure 7 and Figure 8, the process and results of the iterative optimization search for the target parameters under the given constraints are presented. During the optimization, the eigenfrequencies of the operating modes gradually approached the preset target by adjusting the beam width. The final critical beam width ensured that the frequencies of the first-order and second-order modes were approximately 9 kHz, which met the design specifications. The iterative optimization not only achieved the desired modal order but also ensured that the modal frequencies fell accurately within the required range. This process demonstrated the effectiveness of the optimization method for complex structure design, particularly in the flexible control of modal parameters.

The mechanical sensitivity of a gyroscope is influenced by the difference between the intrinsic frequencies of the driving and detecting modes. A smaller frequency difference results in a higher maximum amplitude of the detected vibration, and thus, greater sensitivity. However, as this difference decreases, the bandwidth of the gyroscope also narrows. To maintain optimal performance, the frequency difference between the detecting and driving modes in this study was set to approximately 30 Hz. After iterative optimization, the software produced three sets of optimized parameters, as shown in Figure 9. Validation was then performed to select the appropriate parameters for further simulation.

The optimization results for the structural and electrical parameters are shown in Table 1.

## 3. Simulation Analysis of Vibration Characteristics

### 3.1. Modal Simulation

Modal analysis is critical for determining the eigenfrequencies, vibration shape and vibration stability of a structure, which play significant roles in its design and performance optimization [23]. A finite element model was developed using COMSOL 6.1, followed by a modal simulation of the iteratively optimized Y-axis gyroscope structure. After applying fixed constraints in the solid mechanics module, the first six eigenfrequencies were extracted. The modal vibration shapes and corresponding intrinsic frequencies obtained from the simulation are shown in Figure 10.

First mode: driving mode, where the resonant mass undergoes horizontal resonant motion;

Second-order mode: detection mode, where the intermediate detection mass undergoes out-of-plane resonant motion;

Third-order mode: interference mode, detecting out-of-plane torsion of the mass block around the X-axis;

Fourth-order mode: interference mode, detecting out-of-plane torsion of the mass block around the Y-axis;

Fifth mode: interference mode, driving the frame to undergo out-of-plane resonant motion;

Sixth mode: interference mode, driving the frame to undergo torsional motion around the X-axis.

In structural design, the frequency difference between the operational and interfering modes should be maximized to minimize cross-mode interference. When the intrinsic frequencies of the driving and sensing modes are equal, the gyroscope achieves maximum sensitivity, but this also reduces the bandwidth. Therefore, the driving and sensing frequencies must maintain a certain frequency difference. As shown in Figure 10, the minimum frequency difference between the working mode and the interference mode of the gyroscope structure was 2229.5 Hz, which was greater than 20% of the working mode. Even if the third and fourth modes are excited, the capacitance will not change due to the consistent potential of the left and right detection comb teeth, so it will not affect the working mode. The frequency difference between the driving and sensitive modes was 28 Hz, and the two modes matched well. In the optimization design of gyroscopes, it is necessary to reasonably select the working frequency difference of the gyroscope. On the premise of ensuring the stability of the gyroscope’s working state, a smaller working frequency difference should be designed to achieve greater sensitivity, which is of great significance for improving the performance of the gyroscope [24]. The eigenfrequencies, calculated based on the stiffness and Coriolis mass, show that the driving and sensing mode frequencies were 9374 Hz and 9334 Hz, respectively, with errors of 1.72% and 0.98% compared with the simulation values.

### 3.2. Harmonic Response Analysis

The harmonic response analysis (HRA) method verifies the behavior of a gyroscope at its designed operating frequency, ensuring stable operation within the expected frequency range [25]. As a special time-domain analysis method, HRA is specifically designed to analyze the sustained cyclic response produced by a structural system under cyclic loading. It determines the steady-state response when the load varies according to a simple harmonic law. The primary objective of harmonic response analysis for gyroscopic structures is to calculate the displacement response under electrostatic forces and obtain amplitude–frequency response curves [26]. This analysis ensures that the system avoids resonance points within the operating frequency range. Based on the analysis results, design parameters, such as the structural geometry and support structures, can be adjusted to eliminate unfavorable resonance effects or improve the system stability.

A sweep simulation was conducted to determine the resonator’s response characteristics at various frequency points. The gyroscope structure was observed at the frequency points where resonance peaks occurred, as well as the magnitudes of these peaks. Building on the modal simulation, an electrostatic force and Y-axis rotational angular velocity were applied to simulate the actual working environment. A frequency domain perturbation was introduced in the range of 100 Hz around the structure’s driving frequency for harmonic response analysis, as shown in Figure 11. By observing the X-axis displacement of the driving combs and the Z-axis displacement of the detecting combs, a peak was observed only at 9217 Hz, consistent with the expected results from the modal simulation. No additional peaks were observed near the driving frequency, indicating that the gyroscope demonstrated good stability within this frequency range.

The Q-factor of the gyroscope is an essential parameter that describes the energy loss of the resonator during vibration [27], reflecting energy loss when the system oscillates near the resonance frequency. It directly impacts the gyroscope’s sensitivity, noise level and signal processing effectiveness. Based on the resonance frequency and bandwidth, the quality factors of the drive and detection modes were calculated to be 83,790 and 46,085, respectively. The low energy loss of the gyroscope allows for high sensitivity and a high signal-to-noise ratio, making it suitable for high-precision and high-reliability applications.

### 3.3. Range Simulation Analysis—Mechanical Sensitivity

The range simulation of the gyroscope helped to determine the maximum and minimum angular velocity values for accurate measurement so that the gyroscope can work stably and accurately within the range, as shown in Figure 12; the displacement in the figure represents the response displacement of the detection mode when the angular velocity was input. The analysis of the range simulation curve concluded that the gyroscope had a good linearity within the range of ±600°/s; outside ±600°/s, the gyroscope response began to deviate from the fitted curve. The degree of response of the output signal to the input angular velocity change is the mechanical sensitivity of the gyro, which was calculated according to the fitting curve to be 2.43 × 10^−10^ m/°/s, with a nonlinearity of 3.6 ppm.

In practice, however, the scale factor nonlinearity was affected by many factors, including the nonlinearities in the suspension geometry, parallel plate capacitance, electrostatic stiffness due to the tuned voltage carrier signal, and mode coupling phenomena between the driving and sensing modes [28,29]. All these may have an effect on the scale factor. And the range needs to be decided specifically based on the back-end measurement and control circuitry. Here is just a simulation result in an ideal case, where the purpose was to verify the error between the simulated and calculated values of the scale factor.

At the time of the design, the gyroscope’s Coriolis mass was 1.42 × 10^−8^ kg, according to Formula (5), the mechanical sensitivity of the gyro was 2.14 × 10^−10^ m/°/s. By considering the capacitance parameters of the comb structure, the capacitance sensitivity was determined to be 8.83 × 10⁻^4^ pF/°/s. The difference between the calculated mechanical sensitivity and the simulation result was 11.9%.

## 4. Processing and Characterization

The CMOS-MEMS process is an advanced manufacturing process that integrates MEMS and complementary metal–oxide semiconductor (CMOS) technologies [9,30]. This allows sensors and electronic processing circuits to be integrated on a single silicon wafer, significantly reducing the chip size and assembly costs while enhancing the signal processing efficiency and response speed. The standard CMOS process enables the integration of multiple metals (e.g., ME1–ME6) and dielectric layers. This structure improves the electrode and wiring flexibility and increases the range of the capacitance variation. Deep reactive ion etching (DRIE) is used to fabricate comb electrodes with high aspect ratios and release resonant structures.

The TSMC (Taiwan Semiconductor Manufacturing Company, Hsinchu, Taiwan) 0.18 μm 1P6M technology was used for the CMOS foundry fabrication [31]. The MEMS post-processing was performed on wafers with standard CMOS processes. As shown in Figure 13, PO1 represents the polysilicon layer, and the metal layer was interleaved with the passivation layer deposited on the surface of the silicon substrate. The subsequent MEMS process flow is shown in Figure 14a–f.

First, DRIE local back-engraving was used to thin the resonant structure to a specified thickness, as shown in Figure 15;Metal layer 6 was used to expose the detected straight beam region, and the dielectric layer SiO_2_ was vertically etched using RIE with metal layer 6 as a mask until the silicon surface layer was stopped;The top metal layer ME6 was removed using plasma etching;Isotropic etching was used to etch through the bottom of the silicon structure to release the detected resonant beam structure;A secondary RIE was performed to etch the dielectric layer SiO_2_ to expose regions other than the detection beams, where M5 was used as a mask to accurately define the microstructure of the gyro;DRIE etching was performed again to form a high depth-to-width ratio that overhung the structure composed of single crystal silicon and composite layers.

A step profiler is a contact-based surface-topography-measuring instrument used to accurately determine the step height or surface profile of materials, with a wide range of applications in groove-etching processes. Its working principle involves the stylus making direct contact with the surface of the object being measured. As the stylus slides along the surface, small peaks and valleys cause it to move up and down, reflecting the surface contour. The sensor then converts the stylus displacement into an electrical signal, which is subsequently transformed into a digital signal and analyzed by software to generate detailed surface contour data. As shown in Figure 16, the step gauge was ideal for measuring regular surfaces with unidirectional layouts and is typically used to measure low-hardness material samples, requiring a minimal measurement force to avoid damaging the surface.

In the experiment, this was equivalent to measuring the longitudinal depth. Due to the inability of in-plane testing to determine the depth in all longitudinal directions, i.e., whether the etching effect was consistent in all longitudinal directions, the step meter testing method had certain limitations. According to the test results of the stair step meter, the structural thickness could be obtained, as shown in Table 2.

The gyro structure obtained after the MEMS post-processing based on CMOS standard wafers is shown in Figure 17. The surface of the mass block was flat, without cracks or damage, and the combs and the beam structure were completely etched according to the design, with no structural breaks or deformations, which ensured the subsequent testing of the vibration and the sensitivity of the test. The comb metal and oxide layer were not delaminated to produce warping and other irregular deformations. 

When designing the drive detection comb, in order to ensure the linearity of the drive displacement and capacitance change, a small spacing of 3.5 μm was set to alternate with the large spacing of 4.5 μm. However, the manufacturing results show that the lower side of the drive detection comb was shifted to the left, and the upper side of the comb gap was shifted to the right, with an offset amount of 0.3 μm–0.5 μm, which resulted in the decrease in the difference between the upper side of the large spacing and the small spacing and the increase in the lower side of the large spacing and the small spacing. After the calculation, the change in the drive-detection capacity was much smaller than expected and may have been nonlinear. The deviation of the comb teeth was considered to be due to the rotation of the inner detection mass block by gravity through the detection beam and the outer drive frame, which resulted in a deviation of the comb teeth gap. It is possible to set up corrective combs to suppress the displacement of the detection mass block by gravity. Alternatively, the deflection movement can be suppressed by considering a change in the type of beam.

The CMOS-MEMS process has multiple metal layers for easy wiring. Because metal can be flexibly used as a mask, an appropriate process design can ensure that subsequent processing does not require customizing additional mask plates. Introducing a silicon substrate for structural support can avoid curling and delamination of the structure. In addition, due to the thickness limitation, only five metal layers are used for capacitor driving and detection. The effective capacitance size used for driving is much smaller than that of conventional MEMS gyroscopes, and an insufficient driving force limits the size and quality of the gyroscope. However, in the future, electrode isolation can be established between the comb teeth and the resonant mass, and the comb teeth can be suspended through the joint of the metal oxide structure layer, which can further enhance the driving force of the gyroscope.

## 5. Test

### 5.1. Static Capacitance Test

The capacitance values between the electrodes and the resonator of the MEMS gyro were tested separately in the room temperature environment of the clean room, and the capacitance values in the stationary state were obtained, as shown in Table 3.

In this study, a semiconductor parameter analyzer (Keithley 4200A-SCS semiconductor parameter analyzer, Keysight Technologies, Santa Rosa, CA, USA) was used to detect the capacitance signal, which could be measured in a wide frequency range of 20 Hz to 1 MHz. The static capacitance test operating platform is shown in Figure 18.

After the testing, it was found that there was a certain deviation between the test values and the calculated values. After the analysis, it was considered that the main reason was that the AOE caused damage to the metal layer sidewall during the SiO_2_ etching process, which resulted in an increase in the spacing between the capacitors and a decrease in the capacitance. At the same time, due to the influence of gravity, the movement of the mass block may have caused changes in the gap between the comb teeth, which may have also led to changes in the size of the capacitance.

### 5.2. Modal Response Testing

Modal response testing focuses on testing the frequency and Q-factor of the two operating modes of a gyro. The operating modes of a gyro are usually the resonant vibration characteristics of the drive and detection modes. By applying a sinusoidal excitation (drive force) to the gyro and gradually adjusting the input frequency, the system will resonate and vibrate to its maximum amplitude when the input frequency is close to its intrinsic frequency. The Q-factor is used to measure the energy loss of a gyro-vibration system. A high Q-value indicates that the system has low energy loss, good resonance and a more sensitive system response. By observing the response of the system at the point of resonance and measuring the two frequency points when the amplitude decays by 3 dB from its maximum value, the Q-factor is then calculated and used to assess the energy loss and stability of the vibration system.

To test the MEMS gyroscope’s modal response, a sinusoidal signal was applied to the drive mode. The experimental principle is shown in Figure 19, where the test circuit was the gyro driving circuit, and the test equipment contained a regulated power supply (GPS-2303C, GWINSTEK, Taipei County, Taiwan), a mixed-signal oscilloscope (Tektronix MSO/DPO4000B, Tektronix, Beaverton, OR, USA) and a signal generator (Keysight 33220A, Keysight Technologies, Santa Rosa, CA, USA).

The swept signal output from the generator was fed into the drive feedback signal input test terminal of the test circuit board and loaded onto the drive electrode E_D1_ of the resonant gyro, and the other was fed into the mixed-signal oscilloscope for observation; two signals were output from the sensitive terminals of the drive feedback electrodes E_DS1_ and E_DS2_ of the test circuit board, where one was fed into the digital multimeter to read the output voltage value, and the other was outputted to the mixed-signal oscilloscope for observation. The frequency of the alternating signal was gradually increased from 8000 Hz to 10,000 Hz, the frequency of the applied signal and the amplitude of the voltage detected by the multimeter were recorded, and the frequency of the signal corresponding to the maximum voltage value was the resonant frequency of the gyro drive mode. Applying the same method to the gyro detection electrode E_S2_, the applied frequency gradually increased the alternating voltage, which was amplified by the instrumentation amplifier using a digital multimeter in the detection of the feedback electrode E_SS1_ and E_SS2_ to detect the voltage value and then recorded. The experimental principle is shown in Figure 20.

The MEMS gyroscope was tested with a DC component of 5 V and an AC amplitude of 100 mV. The resulting frequency response curve for both the drive and detection modes is shown in Figure 21.

The resonant frequency of the gyro drive mode was 9043 Hz, and the quality factor of the drive mode was 475.89. From the figure, the resonant frequency of the MEMS gyro drive mode was 9075.9.4 Hz, and the Q-factor of the drive mode was calculated to be 349.03.

According to Table 4, from the modal response test, it was obtained that there was a certain error between the simulation analysis and experimental test values of the resonance frequencies of the driving and detection modes, which was mainly due to the theoretical modeling error of the sensitive structure, the processing error and the material properties. However, these two kinds of errors were fully considered in the material selection and overall structural design, and the test results could still meet the design requirements.

## 6. Conclusions

The single-axis gyro designed in this study used an ISOD-type structure for in-plane driving and out-of-plane detection. The gyro in the study adopted a multi-objective genetic algorithm to optimize the key beams of the resonant structure in order to achieve the frequency optimization of the driving and detecting modes. The modes and eigenfrequencies of the gyro were verified by using finite element analysis software COMSOL 6.1, and the harmonic response simulation was carried out for the gyro, where the simulation results show that, ideally, the gyro operated at 9 kHz, the frequency difference between the driving mode and the detecting mode was at 28 Hz, and the Q-value was 83,790. The mechanical sensitivity of 2.43 × 10^−10^ m/°/s could be obtained from the range simulation, which was 11.9% different from the calculated error. And the gyro had a low nonlinearity of 0.036% at ±600°. However, the nonlinearity of the gyro’s scale factor in the actual working condition was affected by many other factors, such as the nonlinearity of the suspension geometry, parallel plate capacitance, electrostatic stiffness caused by the tuned voltage carrier signal, and the phenomenon of mode coupling between the driving mode and the sensing mode. The gyro was post-processed through the MEMS process on CMOS standard wafers, supported by a silicon substrate to improve the overall stiffness, the completed gyro structure was completely released, and the multi-layer metal stacked comb and beam structure were not fractured and deformed. The gyro structure designed by this process could greatly improve the integration degree to reduce costs. The gyro chip was tested, and its static capacitance value was in error relative to the design value. Considering the influence factor of the capacitance, the error was caused by the damage of AOE etching on the metal sidewall. The frequency characteristics had an error of 1.9% from the simulation results, which was due to the stiffness of the beam caused by the machining error. The Q-values of the driving and detection modes under atmosphere were 475.89 and 349.03, respectively. The gyro will be subsequently tested in a vacuum package to further improve the performance of the gyro, and other performance parameters of the gyro will be tested and investigated. Since the silicon substrate in this study is only used as a support and does not provide any electrical quantities, in order to improve the performance of the gyro, it is considered that the electrode isolation is added in the next step of the design, and the silicon substrate is introduced into the design of the driving electrodes to increase the driving force of the gyro.

## Figures and Tables

**Figure 1 micromachines-15-01484-f001:**
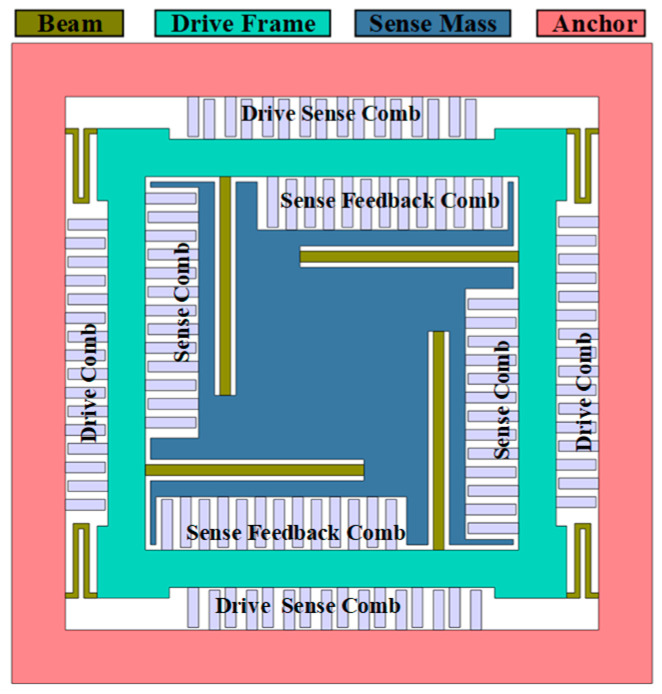
Out-of-plane detection gyro structure design.

**Figure 2 micromachines-15-01484-f002:**
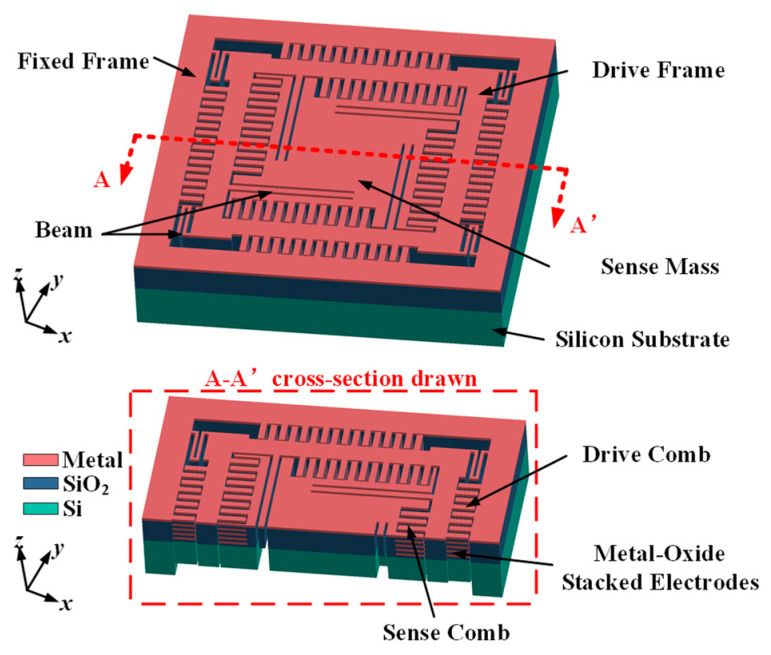
Overall structural design concept.

**Figure 3 micromachines-15-01484-f003:**
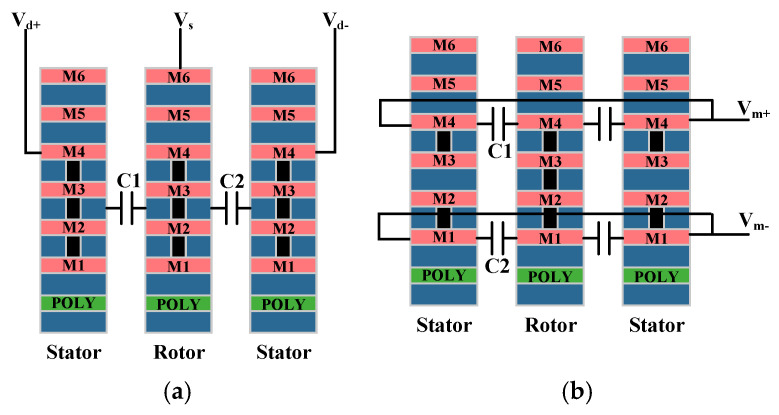
Design of drive and detection combs: (**a**) in-face drive combs; (**b**) out-of-face detection combs.

**Figure 4 micromachines-15-01484-f004:**
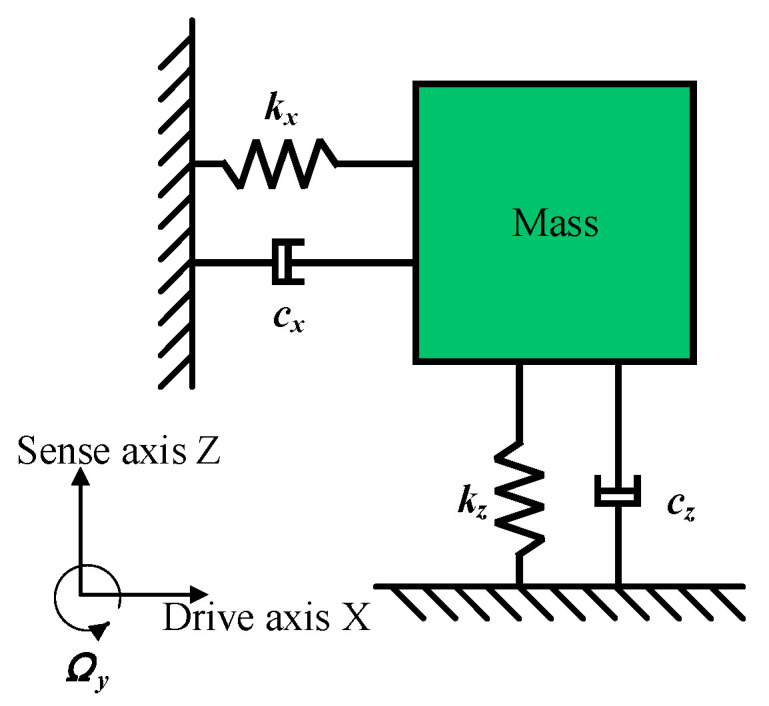
Dynamic modeling of gyroscopes.

**Figure 5 micromachines-15-01484-f005:**
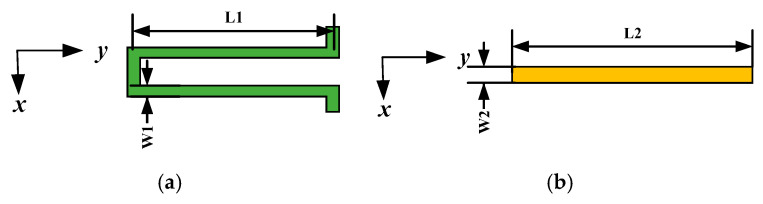
Key beam construction: (**a**) U-shaped beam; (**b**) straight beam.

**Figure 6 micromachines-15-01484-f006:**
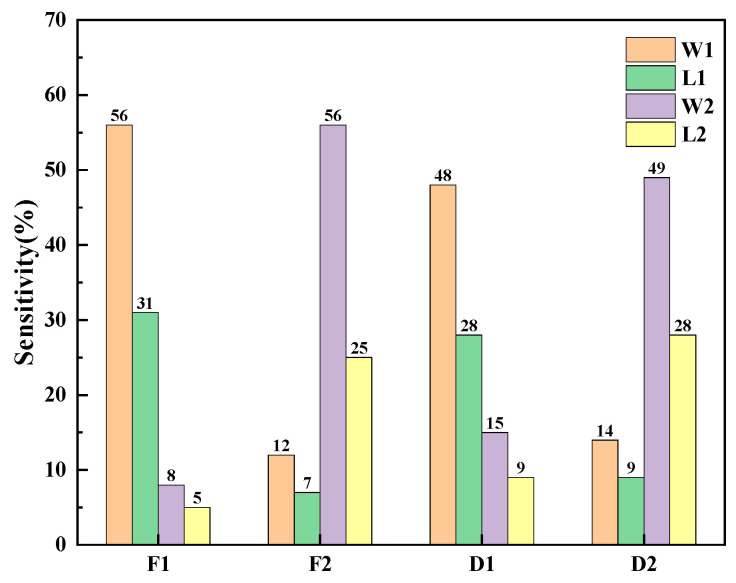
Sensitivity of beams to target parameters.

**Figure 7 micromachines-15-01484-f007:**
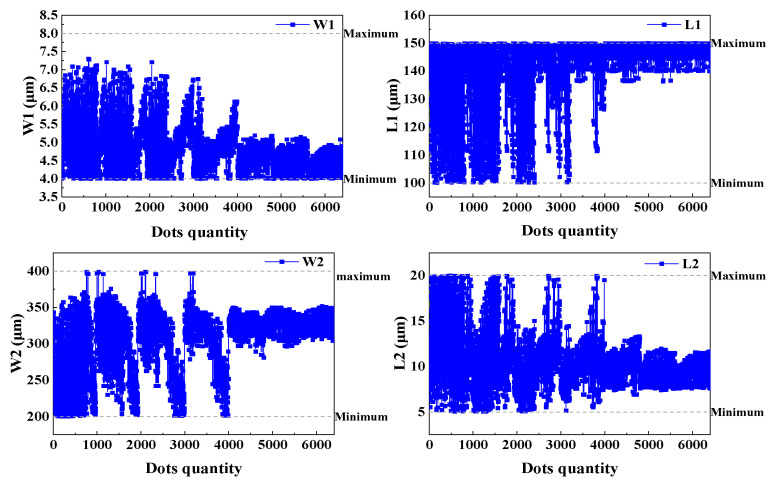
Iteration of beam parameters for the optimization process.

**Figure 8 micromachines-15-01484-f008:**
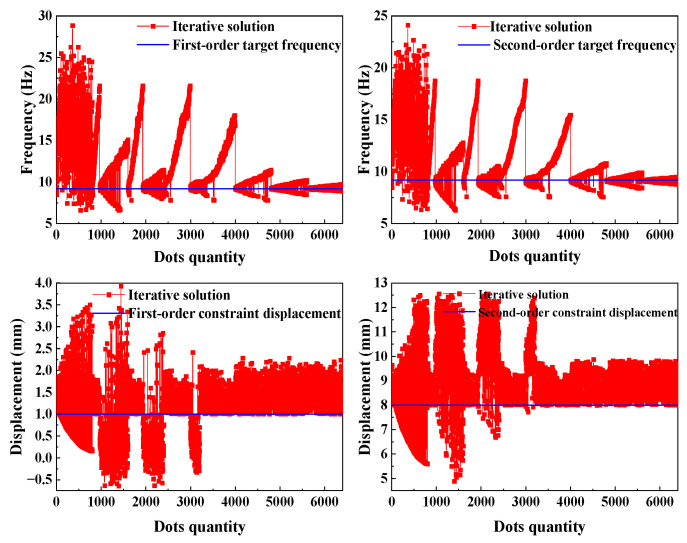
Iteration of the objective parameters of the optimization process.

**Figure 9 micromachines-15-01484-f009:**
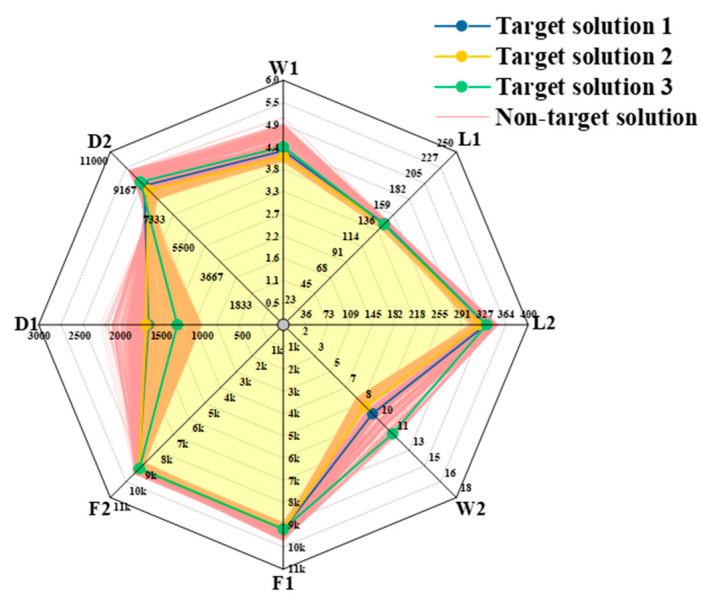
Optimization results.

**Figure 10 micromachines-15-01484-f010:**
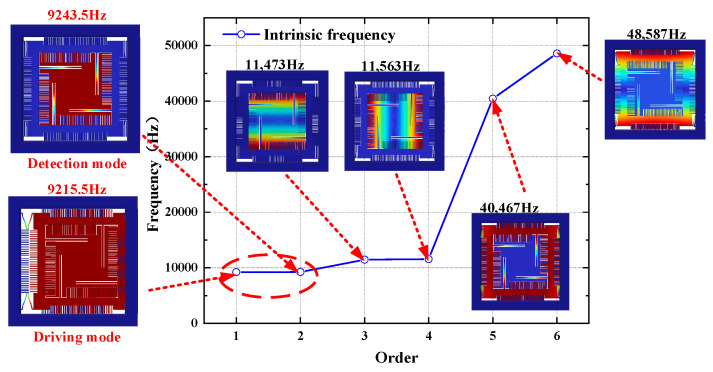
Simulation results of the first six orders of modes.

**Figure 11 micromachines-15-01484-f011:**
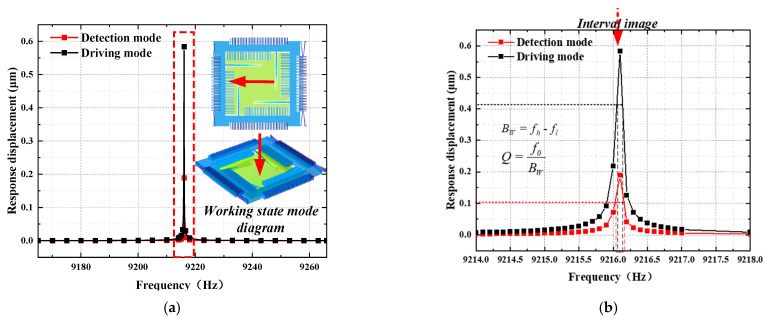
Harmonic response simulation results: (**a**) harmonic response simulation; (**b**) Q-factor calculation.

**Figure 12 micromachines-15-01484-f012:**
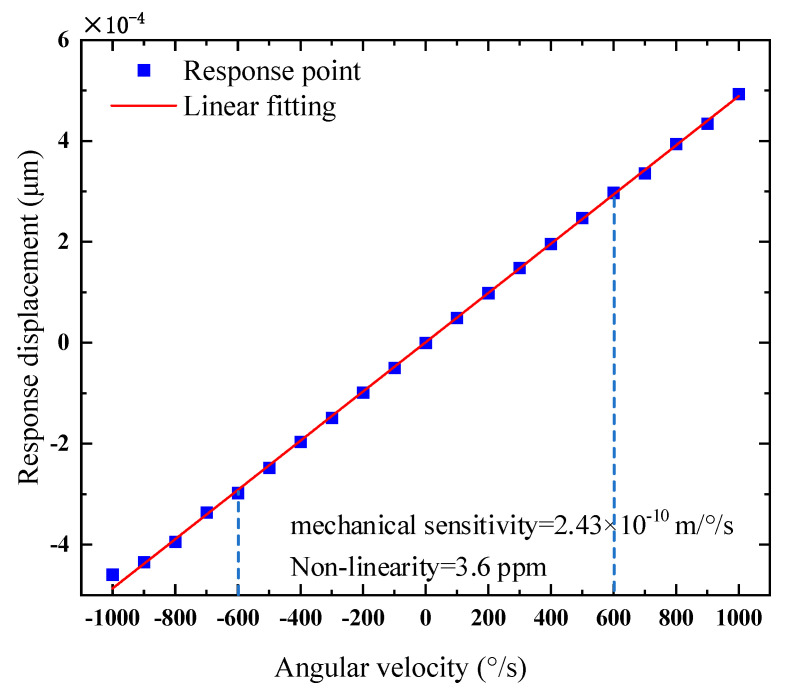
Range and mechanical sensitivity simulation.

**Figure 13 micromachines-15-01484-f013:**
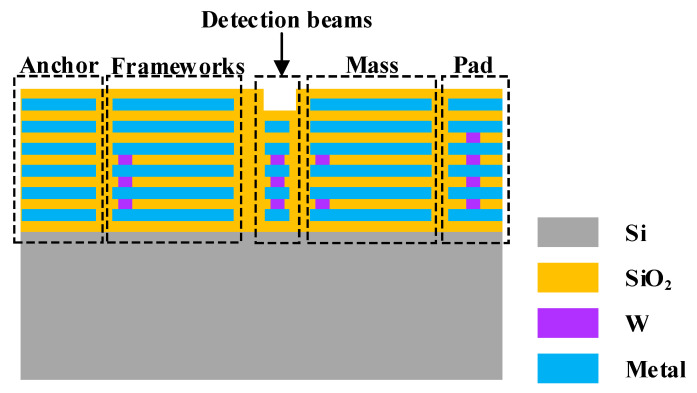
Standard CMOS-MEMS wafers.

**Figure 14 micromachines-15-01484-f014:**
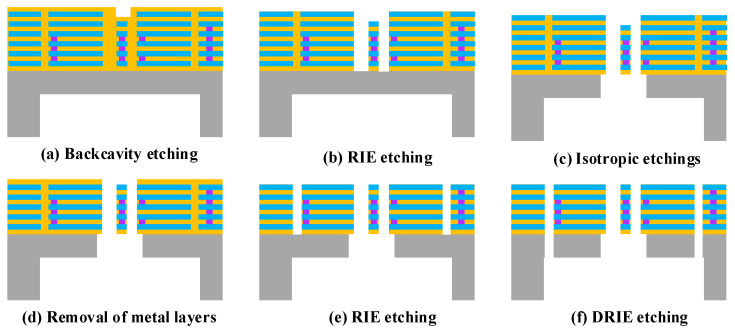
MEMS process flow.

**Figure 15 micromachines-15-01484-f015:**
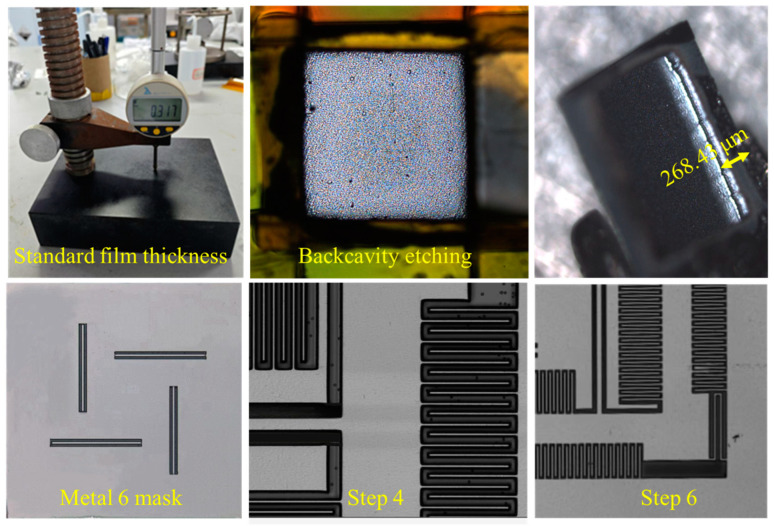
Photographs of processing.

**Figure 16 micromachines-15-01484-f016:**
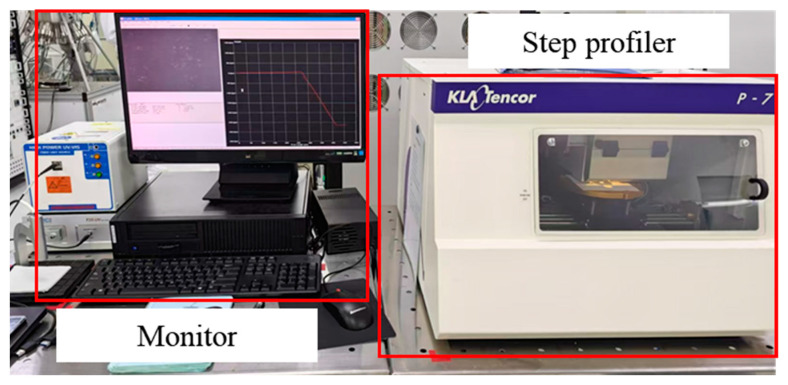
Stair tester testing platform.

**Figure 17 micromachines-15-01484-f017:**
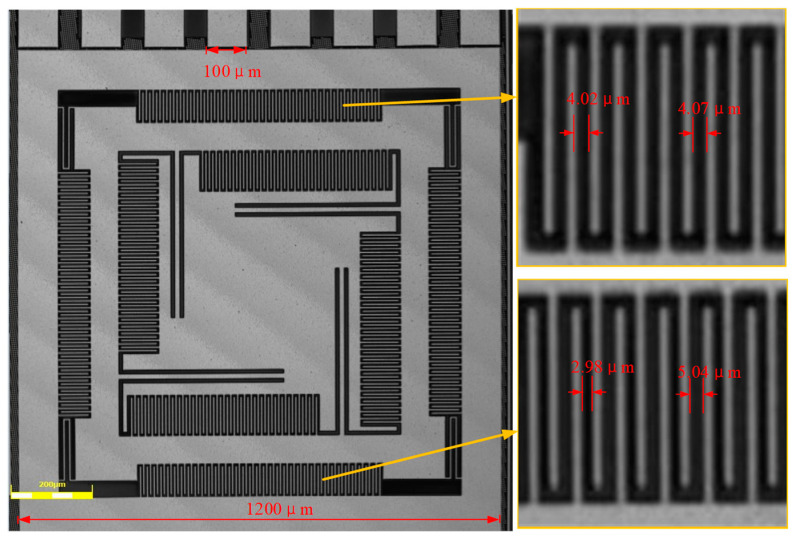
Photographs of processing results.

**Figure 18 micromachines-15-01484-f018:**
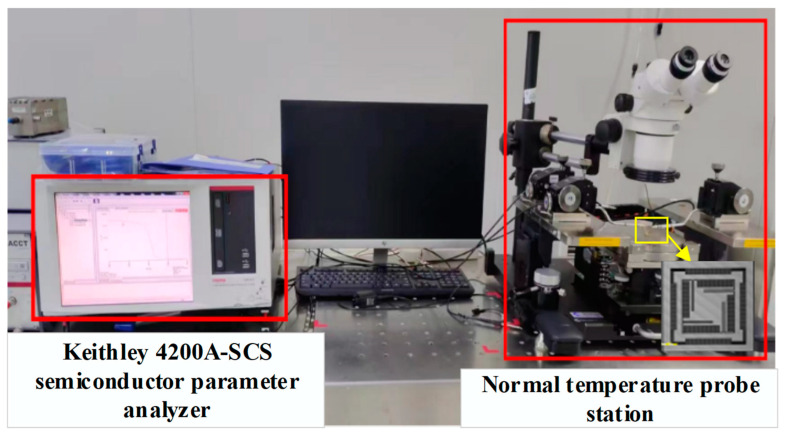
Static capacitance test platform.

**Figure 19 micromachines-15-01484-f019:**
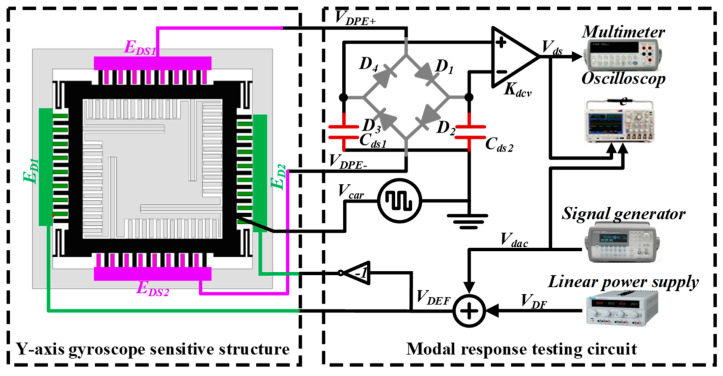
Experimental principles of driven modal response.

**Figure 20 micromachines-15-01484-f020:**
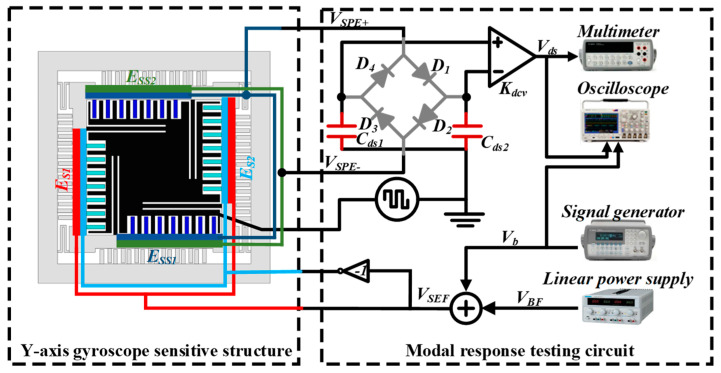
Experimental principle of detecting modal response.

**Figure 21 micromachines-15-01484-f021:**
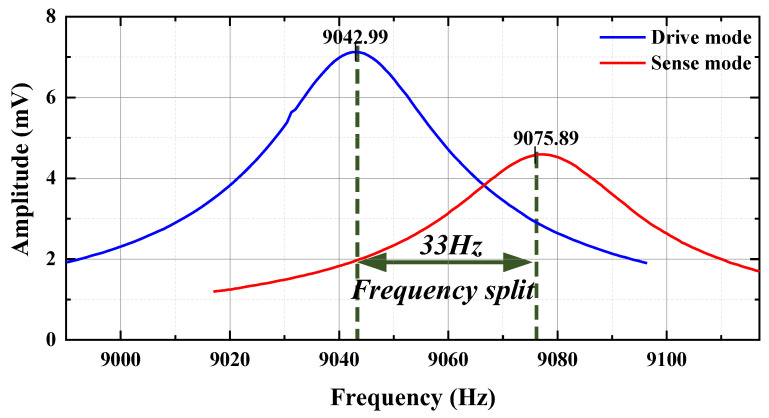
Experimental results of gyro modal response experiment.

**Table 1 micromachines-15-01484-t001:** Results after beam optimization.

Parameter	Value
L1 (μm)	144.94
W1 (μm)	4.37
k_ux_ (N/m)	101.90
L2 (μm)	352.50
W2 (μm)	11.93
k_z_ (N/m)	50.52
Structural thickness (μm)	40
Detecting beams thickness (μm)	6
Capacitive gap (μm)	4
Chip size (μm)	1200 × 1300
Drive static capacitor (F)	3.75 × 10^−14^
Detecting static capacitance (F)	1.62 × 10^−14^

**Table 2 micromachines-15-01484-t002:** Etching thickness result.

Parameters	Dimensions (μm)
Anchor frame thickness	317.0
Resonant structure thickness	40.07
Detecting beam thickness	5.84

**Table 3 micromachines-15-01484-t003:** Static capacitance test results.

Inter-Electrode Capacitance	Calculated Value (F)	Test Value (F)
M-D1	3.75 × 10^−14^	3.24 × 10^−14^
M-D2	3.75 × 10^−14^	3.28 × 10^−14^
M-S1	1.64 × 10^−14^	1.48 × 10^−14^
M-S2	1.64 × 10^−14^	1.51 × 10^−14^
M-DS1	3.76 × 10^−14^	3.31 × 10^−14^
M-DS2	3.76 × 10^−14^	3.27 × 10^−14^
M-SS1	1.64 × 10^−14^	1.47 × 10^−14^
M-SS2	1.64 × 10^−14^	1.49 × 10^−14^

**Table 4 micromachines-15-01484-t004:** Comparison of modal response test values with theoretical values.

Frequency	Drive Mode (Hz)	Detecting Mode (Hz)
Simulation values	9215.5	9243.5
Experimental test values	9043.0	9075.9

## Data Availability

The original contributions presented in this study are included in this article.

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
