# Peer review of "Design and Implementation of a CMOS-MEMS Out-of-Plane Detection Gyroscope"

_micromachines, 2024, doi:10.3390/mi15121484_

Round 1
Reviewer 1 Report (New Reviewer)
Comments and Suggestions for Authors
The authors present their research work about out-of-plane detection gyroscopes in CMOS-MEMS technology, providing deep insights on the design strategy and optimization and preliminary characterization (resonance frequency and static capacitance measurement).
The manuscript clearly describes the theoretical background, methodolody, design and test phase, but sometimes lacks in claryfing the novelty introduced by this work.
Major points:
1) even though there are few examples in literature about CMOS-MEMS out-of-plane detection gyroscopes, the authors should compare their result and design with existing research on the topic to provide a fair comparison and to highlight the inroduced novelty with the state-of-the-art. Some out-of-plane detection CMOS-MEMS gyroscopes in the literature can be found here: DOI 10.1109/MEMSYS.2001.906505 and DOI: 10.1109/IMPACT.2009.5382177 .
2) Some of the figures and related captions should be improved in terms of clarity. E.g. In Figure 6 the parameters F1, F2, D1 and D2 are introduced but they are not described anywhere in the text nor in the caption of the figure (I assume they are the frequencies and displacement for resonant modes 1 and 2), in Figure 12 I assume the response displacement refers to the displacement of the Coriolis mass in the sense direction, but again it is not defined anywhere in the text/caption., etc. I believe that fixing this aspect could really improve the manuscript clarity.
3) Could the authors comment on the 9 kHz resonance frequency target? In section 2.3 the authors states that "the operating frequency of the gyro resonator is minded to be in the range of 5 to 15 kHz", however I do not totally agree on this point as the said frequency range is within the audio range (20 Hz to 20 kHz) and it is usually preferrable to design an inertial sensor with resonance frequency above said range to improve vibration/acoustic rejection of the sensor. Could you achieve similar performance targeting a higher resonance frequency (i.e. >20 kHz)? If not, what is the main limitation?
4) Could the authors comment on the two spurious modes at 11.5 kHz? Again, for vibration rejection performance spurious modes are typically kept at a higher distance (e.g. >5-10 kHz) from the main operational modes, what is the main limitation in terms of mechanical design to achieve a higher operational modes to spurious modes separation? Also, it seems to me that the modes at 11.5 kHz displace the Coriolis mass in the out-of-plane direction (the same directio of the sense mode). Could the authors comment on this point?
5) Figure 8 shows the displacements of the first and second modes, but there may be an error as the displacement are in the range of some millimeters which are higher by orders of magnitude than usual displacement in MEMS sensors (few micrometers to tens of micrometers). Could the authors comment on this point?
6) Table 3 presents the static capacitance test results. It seems that the difference between the theoretical and measured value is pretty constant (measured is 85-90% of the theoretical), thus suggesting for a deterministic root cause behind the ideal to real capacitance difference. Could the authors comment on this point? Did the authors studied which process parameter might cause this difference?
7) Finally, at the end of section 5.2 it is said that the quality factors are 475.89 and 349.03 which is not in line with the values provided in the abstract and conclusion (83790 and 46085). Could the authors comment on this point?
Minor points:
8) line 152 Ay should be Az
9) equation (5) typo in Smechanical
10) line 276 "This design minimizes interference from environmental vibrations and maintains sufficient bandwidth to ensure high sensitivity and achieve optimal dynamic response". I believe this statement is misleading as requests on vibration rejection and bandwidth are highly dependent on the target application, so there is not a true optimal design if the target application is not defined. For example, the split value for the presented gyroscope is 30 Hz which limits the bandwidth to something below 10 Hz, which is not sufficient for certain applications. I suggest to better rephrase it considering this comment.
11) Figure 15 bottom left image 'Matel6' should be corrected with 'Metal6'
12) line 382 'in the following table' should be modified into 'in table 2'
13) line 403 'in Table 2' should be modified into 'in table 3'
Overall, the quality of English language in the presented manuscript does not limit the undestanding of the research work. Anyway, some paragraphs could benefit from careful proof-reading. For example: line 166, line 432-438
Author Response
Please see the attachment.

Reviewer 2 Report (New Reviewer)
Comments and Suggestions for Authors
The manuscript presents an interesting approach to MEMS gyroscope design; however, there are several areas that need substantial improvement to make the work more comprehensible and impactful. Below are the detailed comments and suggestions for improvement.
The introduction lacks depth. While it references other fabrication methods and performances, their significance is unclear. More discussion is needed on their relevance to this work, their limitations, and how this study addresses those gaps. The novelty of the work should be highlighted, especially if it aims to compare or improve upon existing methods. A detailed explanation of the noise floor and its implications for the design is also missing.
Figure 2 has visual issues. Labels with a white background block parts of the structure and should be replaced with transparent or less obtrusive alternatives. Additionally, the location of the anchor is unclear and should be explicitly marked.
Section 2 fails to explain the fundamental concepts adequately. The mechanical mode of the gyroscope is unclear, and a mode shape illustration here is necessary (figure 10 is too late). There’s insufficient detail about how the inner sense mass detects the Coriolis force or how the drive sense comb handles horizontal motion. Missing design parameters include the capacitance gap size, structure thickness, and total drive and sense capacitance. Sections 2.1 and 2.2 are overly basic and should be condensed, retaining only the essential formulas and references to existing works.
Also, this design reduces the size of a MEMS y-axis gyroscope and implements it through the CMOS-MEMS process. A detailed discussion comparing the benefits and drawbacks of this approach is necessary, especially regarding the CMOS-MEMS process, and the challenges and benefits of this approach are not discussed.
The mechanical sensitivity in Section 3 is presented superficially. Suspension nonlinearity, a crucial aspect, is not apparent and it is not clear if it was considered or not. Relevant references for your convenience, such as “Characterization of Scale Factor Nonlinearities in Coriolis Vibratory Gyroscopes” and “Effect of EAM on Capacitive Detection of Motion in MEMS Vibration Gyroscopes.” A lot of papers are included in this discussion.
The sensitivity depends on several factors, and the angular gain, approximately 0.7 for this design, is missing from the equations. Without real device measurements, the claim of a ±600 dps range based on a simulation seems overly optimistic.
The conclusion is too simple and reads like a repetition of the abstract. It should discuss the results in more detail, explaining how they validate the design and addressing limitations. Suggestions for future research would also add value.
Overall, significant revisions are required to improve the paper. This includes expanding the introduction, clarifying key concepts, providing missing design details, addressing errors in the equations, and revising the conclusion for meaningful discussion. These changes will make the study more rigorous and impactful.
Round 2
Reviewer 2 Report (New Reviewer)
Comments and Suggestions for Authors
Thank you for considering my comments and addressing most of them. However, I still strongly recommend simplifying Section 2.2. There is no need to derive those equations again, as they are well-established in the literature. Including them adds unnecessary length to the manuscript without providing additional value.
Regarding Figure 20, the light blue (Es2) appears unused. This might be an error in the drawing and should be corrected.
Additionally, there is a significant discrepancy between the capacitive gap values listed in Table 1 (4 µm) and Figure 17 (0.35 µm). Could you clarify why the fabrication results deviate so much from the design parameters? This needs further explanation.
Lastly, in the drive mode, the arrangement of the drive sense comb electrodes in the figure raises questions. Their current placement suggests they cannot measure horizontal motion effectively. This should be addressed in the paper, as it might be a design flaw. Ideally, these fingers should be arranged as parallel plates with alternating small and large gaps to ensure proper functionality.
Round 3
Reviewer 2 Report (New Reviewer)
Comments and Suggestions for Authors
Thank you for your prompt response. However, I kindly ask you to note that correcting errors should not be done to simply replacing images or text in the article. Such actions are equivalent to altering data and experimental parameters, which is a matter of great seriousness. Please ensure thorough checks are conducted before future publications.
Additionally, while the correction to a parallel-plate capacitor in design is accurate, it is important to clarify that the sensor already fabricated cannot be modified to reflect the correct capacitor type. I suggest including this point in the article along with a discussion.
Author Response
Please see the attachment.

This manuscript is a resubmission of an earlier submission. The following is a list of the peer review reports and author responses from that submission.
Round 1
Reviewer 1 Report
Comments and Suggestions for Authors
This paper presents a CMOS-MEMS out-of-plane detection gyroscope, which can be applied in inertial navigation fields. The topic is within the scope of this journal. The organization of the paper is fine, but there is still something should be improved. Some of my comments are:
1. The author mentioned the Y-axis gyroscope in the article. Why does the theoretical part show the Z-axis angular velocity symbol? Please check and modify the formula. The meaning of each value is not specified in the formula in Chapter 2.2, please supplement.
2. The constraints and optimization objectives used during the iteration process are not clearly stated in the text. It is recommended to add them.
3. The X-axis and Z-axis markings in Figure 4 are reversed. Please make the necessary changes.
4. It is recommended to modify it by changing “detection” to “sense” in figure 1.
5. The unit of non-linearity is irregular, it is recommended to use “ppm”
6. The article did not mention testing, so I think the abstract section 'and testing' should be deleted
Comments on the Quality of English LanguageThe English should be further polished, some sentences should be written in a better format.
Reviewer 2 Report
Comments and Suggestions for Authors
This manuscript presents the design, simulation, fabrication, and testing of a Y-axis gyroscope using CMOS-MEMS technology. The work highlights the optimization of the gyroscope structure through a multi-objective genetic algorithm and analyzes the performance through modal and harmonic simulations. The CMOS-MEMS approach provides integration and cost advantages, which are pertinent to the advancement of micro-gyroscope technologies.
However, the manuscript requires additional validation to strengthen its scientific impact and practical relevance. The manuscript currently emphasizes simulation data without a thorough experimental verification of the gyroscope's performance. Including detailed experimental results, such as measurements of sensitivity, Q factor, and frequency response in actual use conditions, would enhance the robustness of the claims. To improve the manuscript's impact, the authors should compare the measured performance metrics of this gyroscope with similar designs or commercially available gyroscopes.
Round 2
Reviewer 2 Report
Comments and Suggestions for Authors
I agree with the acceptance of the revised manuscript now.